# Effects of bright light therapy for depression during pregnancy: a randomised, double-blind controlled trial

Babette Bais  ,[1] Astrid M Kamperman,[1] Hilmar H Bijma,[2] Witte JG Hoogendijk,[1] Jan L Souman,[3] Esther Knijff,[1] Mijke P Lambregtse-van den Berg[1,4]

[1]Psychiatry, Erasmus Medical Center, Rotterdam, Zuid-Holland, The Netherlands
[2]Obstetrics and Gynaecology, Erasmus Medical Center, Rotterdam, Zuid-Holland, The Netherlands
[3]Lighting Applications, Signify NV, Eindhoven, Noord-Brabant, The Netherlands
[4]Child and Adolescent Psychiatry/Psychology, Erasmus Medical Center, Rotterdam, Zuid-Holland, The Netherlands

**Correspondence to**
Dr Babette Bais;
b.bais@erasmusmc.nl

## ABSTRACT

**Objectives** Approximately 11%–13% of pregnant women suffer from depression. Bright light therapy (BLT) is a promising treatment, combining direct availability, sufficient efficacy, low costs and high safety for both mother and child. Here, we examined the effects of BLT on depression during pregnancy.

**Design** Randomised, double-blind controlled trial.

**Setting** Primary and secondary care in The Netherlands, from November 2016 to March 2019.

**Participants** 67 pregnant women (12–32 weeks gestational age) with a DSM-5 diagnosis of depressive disorder (Diagnostic and Statistical Manual of Mental Disorders).

**Interventions** Participants were randomly allocated to treatment with either BLT (9000 lux, 5000 K) or dim red light therapy (DRLT, 100 lux, 2700 K), which is considered placebo. For 6 weeks, both groups were treated daily at home for 30 min on awakening. Follow-up took place weekly during the intervention, after 6 weeks of therapy, 3 and 10 weeks after treatment and 2 months postpartum.

**Primary and secondary outcome measures** Depressive symptoms were measured primarily with the Structured Interview Guide for the Hamilton Depression Scale— Seasonal Affective Disorder. Secondary measures were the Hamilton Rating Scale for Depression and the Edinburgh Postnatal Depression Scale. Changes in rating scale scores of these questionnaires over time were analysed using generalised linear mixed models.

**Results** Median depression scores decreased by 40.6%– 53.1% in the BLT group and by 50.9%–66.7% in the DRLT group. We found no statistically significant difference in symptom change scores between BLT and DRLT. Sensitivity and post-hoc analyses did not change our findings.

**Conclusions** Depressive symptoms of pregnant women with depression improved in both treatment arms. More research is necessary to determine whether these responses represent true treatment effects, non-specific treatment responses, placebo effects or a combination hereof.

**Trial registration number** NTR5476.

## INTRODUCTION

Antepartum depression is a common and high impact disease, with approximately

### Strengths and limitations of this study

► We conducted various follow-up measurements, including postpartum, to study the effects of withdrawal of treatment and to study whether treatment during pregnancy would protect against postpartum depression.
► The setting of treatment was within a real-world setting.
► A strength of this study was the comprehensive assessment of side effects, as well as acceptability and satisfaction of treatment.
► An unforeseen lack of resources prevented us from including 150 participants, as we aimed to do according to our sample size calculation.
► Depressive symptoms during the study are assessed by questionnaires, rather than diagnostic criteria.

11%–13% of pregnant women suffering from depression.[1] Antepartum depression is not only seen in autumn and winter, but is a year-round phenomenon, with certain subgroups even showing more symptoms in summer.[2] Many risk factors for antepartum depression have been identified.[3 4] Possible causes for antepartum depression may include alterations in endocrine systems, such as the hypothalamus–pituitary–adrenal gland (HPA) axis,[5] and inflammation.[6 7] Women who suffer from antepartum depression are more likely to suffer from postpartum depression as well.[8] Children who are exposed to maternal depression during pregnancy have a higher risk of adverse birth outcomes, such as prematurity and being small for gestational age.[9 10] Additionally, children of mothers with antepartum depression show more often cognitive, emotional and behavioural problems in childhood, adolescence and adulthood[11 12] and they have a higher risk of suffering from depression later in life.[13] During pregnancy, fetal programming of the HPA axis takes

place, which can be affected by maternal depression during pregnancy and may have long-lasting effects on stress response.[14] Possible mechanisms are (1) maternal cortisol crossing the placenta and thus increasing fetal cortisol levels, (2) placental secretion of corticotropin-releasing factor, which stimulates both maternal and fetal cortisol and (3) reduced blood flow to the fetus, causing fetal growth restriction.[9 15–18] In addition, epigenetic programming takes place within the antepartum period, which influences not only the health of the (unborn) infant, but also that of following generations.[19] Therefore, early detection and treatment of antepartum depression is highly important for both mother and infant.

In non-pregnant women, guidelines propose psychotherapy, antidepressant medication or a combination of both as treatment. However, psychotherapy might not be readily available and the safety of maternal use of antidepressants, which cross the placenta, still remains to be established. The use of antidepressants is controversial, because of potential teratogenicity.[20 21] For example, increased risks have been found for persistent pulmonary hypertension of the neonate[22] and cardiovascular malformations.[23] Furthermore, pregnant women express a strong preference for non-pharmacological treatment because of the possible harm for their unborn child.[24 25] Moreover, current adherence to national guidelines by midwives and gynaecologists is low[26] and international guidelines on the pharmacological treatment of antepartum depression are not consistent,[27] which might result in unwanted variation in practice. Despite this, antidepressant use during pregnancy is increasing, not only in the Netherlands,[28 29] but in other European countries and the USA as well.[30–32] In the Netherlands, approximately 2%–3% of pregnant women use antidepressants.[29 33 34] In the USA, this prevalence is approximately 6%–7%,[35–37] but could even be as high as 15% in some states.[38] Therefore, it is urgent and clinically relevant to investigate alternative approaches to treat antepartum depression, such as bright light therapy (BLT).[39]

Light synchronises the suprachiasmatic nucleus (SCN), or the 'biological clock', with the environmental day–night rhythm.[40] Light hits the retina and intrinsically photosensitive retinal ganglion cells (ipRGCs) in the retina project, via the retino-hypthalamic tract to the SCN and thus influences circadian rhythm,[40–42] which may indirectly benefit depressive symptoms.[43] However, not only do ipRGCs project to the SCN, but also directly to brain regions important in the regulation of mood, such as the medial amygdala and the lateral habenula.[40–42]

Although BLT is the first-choice treatment for seasonal affective disorder, a condition of reoccurring depressions during fall and winter, with remissions in spring and summer,[44 45] the effects of BLT have been shown both in seasonal affective disorder and in non-seasonal depression, which is not only shown by a Cochrane review,[46] but also by more recent systematic reviews and meta-analyses.[47–50] An open trial of BLT in pregnant women showed improvement of mean depression ratings by 49%.[51] Two small randomised controlled trials (RCTs) showed significant improvement of depression among pregnant women exposed to BLT compared with placebo.[52 53] Although these results seem promising, the sample sizes of these studies were small, making them at risk for chance-findings.[54]

In this study, we compared the effectiveness of BLT compared with placebo light among pregnant women with a depressive disorder in a larger randomised clinical trial. Moreover, we followed women until the postpartum period, to study whether treatment with light therapy during pregnancy might protect against postpartum depression. We hypothesised that daily treatment with 6 weeks of morning BLT will improve depressive symptoms during pregnancy.

## MATERIAL AND METHODS
### Design
This study was a randomised, double-blind, placebo-controlled clinical trial (Bright Up, NTR5476, http://www.trialregister.nl). A detailed protocol can be found elsewhere.[55] In short, the aim of the Bright Up Study was to evaluate the effectiveness of BLT for pregnant women with a depressive disorder, compared with placebo light.

### Participants
Eligible participants were pregnant women (12–32 weeks of gestational age, confirmed by ultrasound) diagnosed with a depressive disorder, confirmed by a structured clinical interview for DSM disorders (SCID) by one trained assessor (Diagnostic and Statistical Manual of Mental Disorders).[56] The specific inclusion and exclusion criteria are listed in table 1.

In the earlier published study protocol,[55] we aimed to include women who were 12–18 weeks pregnant. For pragmatic reasons, in particular the fact that a substantial number of women were referred after 18 weeks of pregnancy, we later decided to widen our inclusion criteria to 12–32 weeks pregnancy.

In the Netherlands, maternity care for low-risk pregnancies is provided by midwives (primary care). High-risk pregnancies are cared for by gynaecologists in a general hospital (secondary care) or fetal–maternal medicine unit (tertiary care).

In this study, women were recruited not only via healthcare professionals, such as general practitioners, midwives, gynaecologists, psychiatrists and psychologists, but also via (social) media. A complete flow-chart of the recruitment can be found in figure 1.

Initially, we calculated the number of women to be included, based on the results and research methodology of previous studies.[51 52 57] We expected a true treatment effect in the range of a 10%–15% symptom reduction over the full course of treatment (six weekly assessments), reflecting a small to medium effect size. A sample size calculation was performed using GLIMMPSE V.2.1.5. software,[58] with the following parameters: alpha 0.05; beta

**Table 1** Inclusion and exclusion criteria for the Bright Up Study

| | Women |
|---|---|
| Inclusion criteria | 18–45 years of age |
| | 12–32 weeks pregnant (as confirmed by ultrasound) |
| | Current DSM-5 diagnosis of depressive disorder (as assessed by the SCID) |
| Exclusion criteria | Insufficient proficiency in Dutch or English |
| | Multiple pregnancy |
| | Current use of antidepressants shorter than 2 months |
| | Lifetime diagnosis of bipolar I or II disorder |
| | Any psychotic episode |
| | Current substance abuse |
| | Current primary anxiety disorder |
| | Recent history of suicide attempt |
| | Current shift-work |
| | Somatic and/or obstetric conditions that override study participation |
| | Previous treatment with BLT |
| | Eye condition (macular degeneration, eye diseases, recent eye surgery) |

BLT, bright light therapy; DSM, Diagnostic and Statistical Manual of Mental Disorders; SCID, structured clinical interview for DSM disorders.

0.80; six time assessments (continuous, equally spaced); primary test: time*treatment interaction; Structured Interview Guide for the Hamilton Depression Scale—Seasonal Affective Disorder version (SIGH-SAD) Scores assumed at baseline: M: 28.0 and SD: 7.0, with a linear decrease in symptom scores up to a mean score of 24.0 in the BLT condition. No symptom change was assumed for the dim red light therapy (DRLT) condition; Hotelling-Lawley Trace correction; base correlation 0.4; decay rate 0.05; no additional scaling factors included.

To demonstrate this a total sample size of 126 participants, 63 per arm was needed. To account for loss to follow-up during and after treatment, we aimed at including 150 women. Inclusion took place in The Netherlands and started on 9 November 2016 and lasted until 15 March 2019. By then, 67 women were included. However, due to limiting resources, we decided to stop the inclusion.

## Patient and public involvement
No patients involved.

## Blinding
Participants were blinded to allocation. Participants were informed that the study aimed to investigate the efficacy of different light colours. They were not informed that one treatment arm was considered placebo treatment. This was in accordance with approval of the medical ethical committee.

Outcome assessors were blinded to the allocation of the participants. Participants were asked not to share any details regarding their treatment towards the assessors. When blinding was broken, the assessor was replaced. The researcher performing the primary statistical analyses (AMK) was blinded to the allocation. The field researcher (BB) was not blinded to the allocation for practical reasons. This researcher made sure lamps of the correct allocation were delivered to the participants. Also, this researcher asked participants about any side effects, keeping the independent assessors blinded to any adverse effects that might break the blinding, for example, strained eyes, and answered any questions from the participants regarding their lamps.

At baseline, we asked about any expectations concerning the treatment with regards to their depressive symptoms. Women could choose whether they expected a negative effect, a small negative effect, no effect, a small positive effect or a positive effect. After the intervention period, the participants were asked whether they were aware of their allocation.

## Light therapy
Light treatment consisted of either active BLT (9000 lux, colour temperature 5000 K) or DRLT (100 lux, colour temperature 2700 K). The photobiological characterisations of these treatments are shown in online supplemental table 1. The original lamps were adjusted in the factory where these are produced (EnergyUp HF3419/01, Philips, Eindhoven, The Netherlands). To ensure that participants are exposed to the same light intensity, the output of the lamps was fixed. For the control condition, the standard LED's in the lamp were replaced by LED's with a lower intensity and a different colour temperature. The lamps in the control condition were positioned at the same distance from the participant as in the experimental condition.

The active light therapy was shown to be effective in other studies.[52 53 57 59] DRLT can be considered to be biologically inactive and thus as placebo treatment.[46] In line with two previous RCTs among pregnant women, we chose 6 weeks of daily light exposure.[52 53]

The lamps were delivered at the participants' home by one researcher (BB) who was not blinded to the allocation of the participants. This researcher did not share anything about the allocation with the participants. After delivery of the lamps and instructions, participants commenced their daily treatment with light for 30 min within 30 min of habitual wake up time for a 6 weeks period. This took place at the participants' home. Participants sat in front of two lamps with a distance of approximately 40 cm (15.8 inches). They received a plastic ruler of this length to ensure of the correct distance. The light boxes were placed in a custom-made scaffolding, so that

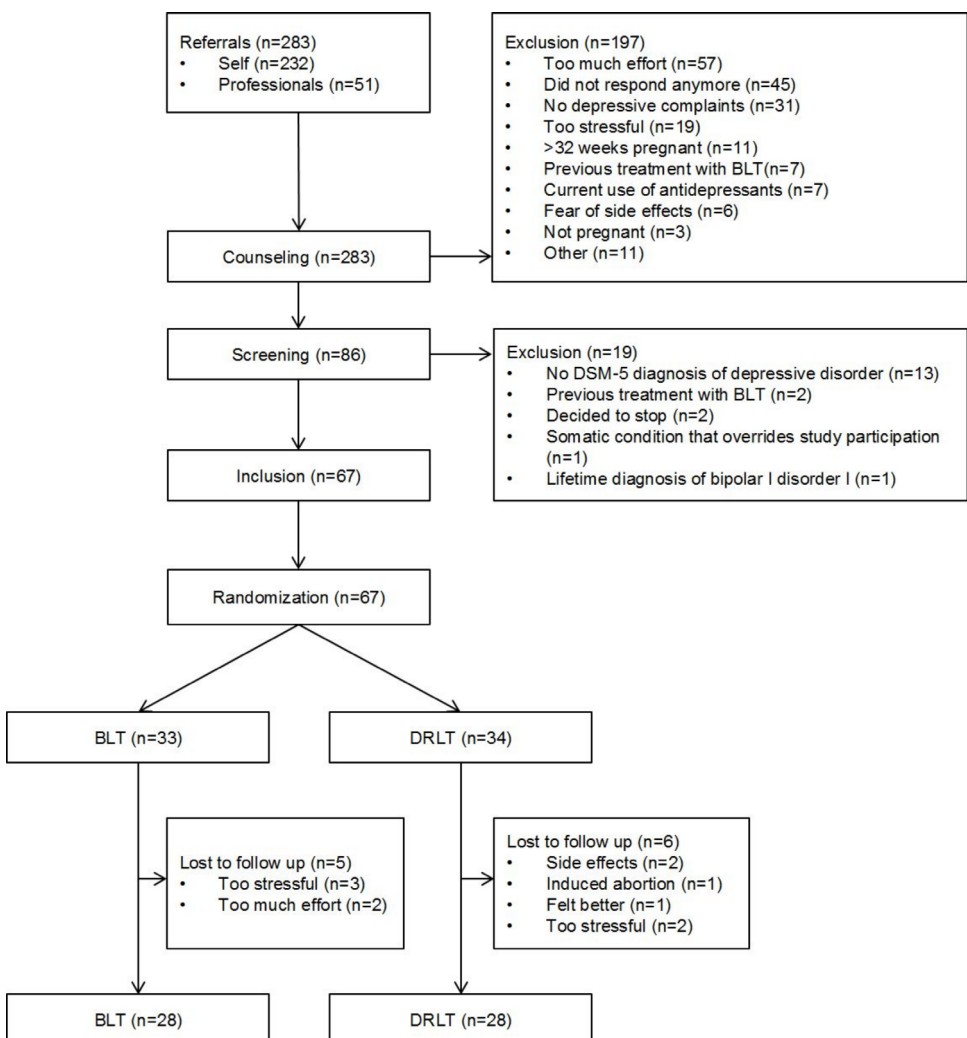

**Figure 1** Flow-chart of the Bright Up Study. BLT, bright light therapy; DRLT, dim red light therapy; DSM, Diagnostic and Statistical Manual of Mental Disorders.

the height of the light boxes could be adjusted per person and glare was avoided. Apart from the light treatment, participants in both treatment arms received treatment as usual: women were free to visit their general practitioner, obstetric care provider or mental healthcare worker and start additional treatment, whenever they felt a need for this.

During the intervention period, self-reported compliance with the light treatment was checked weekly.

## Method

A baseline interview was conducted by telephone by one researcher (BB). The baseline interview collected socio-demographic information (age, ethnicity, educational level, marital status, body mass index (BMI)), obstetric information (gestational age, whether the pregnancy was planned, parity), psychiatric information (substance use (smoking, alcohol, drugs), present and past medication use, present depressive symptoms, psychiatric history) and information on somatic conditions. Also, participants were screened with the SCID for depressive disorder and various potential comorbidities, such as generalised

anxiety disorder and panic disorder. Previous depressive episodes were also assessed with the SCID. The general practitioner was contacted to verify present medication use and whether the participant met any exclusion criteria.

After baseline measurements and receiving written informed consent, the participants were randomly allocated to either receive BLT or DRLT in a 1:1 ratio. Randomisation was done with the web-based computer-generated schedule ALEA (software for randomisation in clinical trials, V.2.2) using random block sizes of 2–6[60] by an independent researcher. Stratification factors were the use of any current antidepressant medication and the number of previous depressive episodes. The latter was dichotomised to three or less versus four or more.[61]

Follow-up took place at the following time points: weekly during the intervention period (T0+1, T0+2 and so on), after 6 weeks of treatment (T1), 3 weeks after end of treatment (T2), 10 weeks after end of treatment (T3), 2 months postpartum (P1), 6 months postpartum (P2), 18 months postpartum (P3).

At these time points, questionnaires were assessed and body material was collected. We collected urine, hair and saliva from the participants, as can be found in our earlier published protocol.[55]

This paper reports the short-term effectiveness, that is, up to 2 months postpartum.

### Primary and secondary outcome measures

The primary outcome measure was the average change in depressive symptoms between the two groups, as measured by the SIGH-SAD. Secondary outcome measures were these changes as measured by the 17-item Hamilton Rating Scale for Depression (HAM-D) and the Edinburgh Postnatal Depression Scale (EPDS).

In the earlier published protocol,[55] we were primarily interested in the effects of light therapy on depressive symptoms. Secondarily, we were interested in the effects on various other outcomes, such as maternal hormonal levels, maternal sleep quality and infant outcomes. Depressive symptoms were measured by two questionnaires: the SIGH-SAD and the EPDS, with the original 17-item HAM-D being part of the SIGH-SAD, which consists of 21 HAM-D items and 8 atypical items. Therefore, in the original protocol,[55] we mentioned these two questionnaires together as the primary outcome, as opposed to the other outcomes (maternal hormonal levels and others). However, it is not technically possible to have more than one primary outcome. Our power calculation was based on the SIGH-SAD, which makes this our true primary outcome. The HAM-D and the EPDS are the secondary outcomes for this manuscript. In the current manuscript, we only report our findings regarding the depressive symptoms. We will report the other outcomes elsewhere. Second, in the trial register, we mention the HAM-D and EPDS as primary outcome, which has been a mistake. The mix-up results from the fact that the SIGH-SAD is in fact the original 17-item HAM-D with an additional four HAM-D and eight atypical depressive items,[62] and the inclusion of women with antepartum depressive mood disorder instead of seasonal affective disorder.

The SIGH-SAD is a 29-item structured interview, consisting of 21 HAM-D items and 8 atypical items. We used the entire SIGH-SAD questionnaire as primary measure, since this is the current benchmark for assessment of depression severity in light therapy trials.[63] We chose the original 17-item HAM-D questionnaire as a secondary measure, since it is more commonly used in clinical practice and research. Blinded assessors conducted the SIGH-SAD interviews (including HAM-D questions) by telephone weekly in the intervention period and at follow-up.

The EPDS is a structured 10-item questionnaire and was used as a self-report measure of depression during pregnancy and postpartum.[64] Items are scored with a value 0–3, resulting in a sum score of 0–30.[64] The EPDS was developed for the detection of postpartum depression, but has been validated for screening depression during pregnancy as well.[65] The EPDS was assessed weekly in the intervention period and at follow-up. Participants received a link by email to fill out the questionnaire.

### Side effects, acceptability and satisfaction

During the intervention period, participants were asked weekly about any possible side effects. Acceptability was assessed by asking participants about their subjective treatment experiences after the intervention period. Women could choose whether they experienced a negative effect, a small negative effect, no effect, a small positive effect or a positive effect. Women were asked how easy or difficult they could implement the therapy in their daily schedule and how easy or difficult the lamp was in use: very difficult, difficult, neutral, easy or very easy. Women could answer whether they found the light therapy very unpleasant, unpleasant, neutral, pleasant or very pleasant. Women were asked whether they would like to use the light therapy outside of the study (yes/no). Finally, women were asked how likely they would recommend light therapy to others on a scale of 1–10.

### Baseline characteristics

The baseline interview collected information on various potential confounders, such as sociodemographic, obstetric and psychiatric information, and information on somatic conditions (see Method for further specifications).

The participant's chronotype was assessed at inclusion with the Munich Chronotype Questionnaire, a structured 19-item self-report questionnaire,[66] since evening types are more prone to depression compared with morning types.[67 68] The participant can be classified into one of seven chronotypes: extremely, moderately or slightly early, normal or slightly, moderately or extremely late. Sum scores range from 16 to 86, with low scores indicating extremely late chronotypes.

### Statistical analysis

Continuous participant characteristics were summarised using mean and SD. Categorical variables, such as educational level, were summarised by count and per cent. In line with the Consolidated Standards of Reporting Trials statement, baseline differences between the two treatment arms were not tested.[69]

For treatment effect analyses, we applied an intention-to-treat procedure, since none of the participants could switch to a different condition, and we included all observations of all participants until the study ended or the participant(s) dropped out of the study.

The primary outcome was changes in SIGH-SAD rating scale scores over time. Secondary outcomes were changes in HAM-D and EPDS rating scale scores over time. Analyses were conducted using general linear mixed modelling analyses. In a series of random-intercept models, we included time (continuous), allocation and time×allocation interaction term as an effect measure of allocation on the course of depression rating scale scores. The standardised baseline score was included in the model, since

baseline depression severity is an important predictor for treatment outcome.[70] We studied the treatment effect for both the intervention period and follow-up period (2 months postpartum).

Primary analyses were first crude, then adjusted. As adjusted primary analyses, we calculated propensity scores based on patient characteristics (psychiatric history, ethnicity, level of education, an unplanned pregnancy, maternal age, parity, gestational age, duration of actual depression and other psychiatric or psychotherapeutic treatment interventions). Next, we adjusted separately for chronotype and the month of treatment. By means of sensitivity analyses, we repeated the primary analyses with last observation carried forward data imputation. As post-hoc analyses, we repeated the crude analyses for women with good compliance (<7 missed treatments) and for women with most severe depressive symptomatology (based on median split baseline SIGH-SAD Scores). Effect parameters were supplied with a 95% CI.

Additionally, we tested responders versus non-responders with Fisher's exact test, where response was defined as a ≥50% decrease to a final score of ≤8 on the 17-item HAM-D and ≤5 on the EPDS at the end of the intervention period.

Data were analysed using SPSS V.21.0 (IBM Corporation). Statistical significance was defined as $p < 0.05$.

## RESULTS
### Demographic and clinical characteristics

In total, 283 women were referred to the study. The majority of the participants (82%) were recruited via (social) media. Of these referrals, we included and randomised 67 women, with 33 allocated to BLT and 34 to DRLT. In total, 11 women dropped out during the study, of whom 5 in the BLT group. Ten women dropped out in the intervention period, one at 10 weeks after treatment. Figure 1 shows a flow-chart of the entire study sample.

Table 2 shows the participant characteristics at the time of inclusion. At inclusion, the mean (SD) of the SIGH-SAD was 26.5 (7.2), of the 17-item HAM-D was 16.9 (5.3) and of the EPDS was 16.1 (4.8). Median scores were respectively 27, 17 and 16. The most common comorbidity was anxiety (25.4%), followed by obsessive compulsive disorder (17.9%), posttraumatic stress disorder (11.9%) and social phobia (11.9%). Various somatic comorbidities were reported, such as asthma, Guillain-Barré syndrome and fibromyalgia.

During the course of this study, as part of the care as usual, 11 additional women started with psychotherapy: three women in the intervention period, one after the intervention period during pregnancy and seven in the postpartum period. During the entire study, four additional women started with psychotropic medication: one woman started with a selective serotonin reuptake inhibitor (SSRI) in the intervention period and one woman in the postpartum period (both sertraline), one with an antipsychotic (quetiapine) and one with a benzodiazepine

(temazepam) postpartum. Of one participant, the dose of the SSRI was increased in the postpartum period (escitalopram).

### Compliance

Self-reported compliance was somewhat higher in the BLT group, compared with the DRLT group. Among the

**Table 2** Overview of participant characteristics at inclusion

|  | BLT (n=33) | DRLT (n=34) |
|---|---|---|
| Age (years), mean (SD) | 31.9 (4.4) | 31.9 (5.3) |
| Gestational age (weeks), mean (SD) | 20.6 (6.2) | 19.7 (6.3) |
| **Ethnicity** | | |
| Dutch | 27 (81.8%) | 26 (76.5%) |
| Other | 6 (19.2%) | 8 (33.5%) |
| **Marital status** | | |
| Married or cohabiting | 33 (100%) | 32 (94.1%) |
| Committed relationship, not cohabiting | 0 (0%) | 1 (2.9%) |
| Single | 0 (0%) | 1 (2.9%) |
| **Education** | | |
| Elementary or (pre-)vocational education | 11 (33.3%) | 13 (38.2%) |
| Higher professional education | 8 (24.2%) | 11 (32.4%) |
| (Pre-)academic education | 14 (42.4%) | 10 (29.4%) |
| **Parity** | | |
| Nulliparous | 15 (45.5%) | 20 (58.8%) |
| Primiparous | 13 (39.4%) | 9 (26.5%) |
| Multiparous | 5 (15.2%) | 5 (14.7%) |
| BMI (kg/m$^2$ or st/ft$^2$), mean (SD) | 25.5 (4.5) | 26.3 (5.4) |
| Planned pregnancy | 22 (66.7%) | 22 (64.7%) |
| Antidepressant medication | 3 (9.1%) | 5 (14.7%) |
| Sleep medication | 3 (9.1%) | 2 (5.9%) |
| Psychotherapy | 14 (48.5%) | 16 (47.1%) |
| **Comorbidities** | | |
| 0 | 17 (51.5%) | 13 (38.2%) |
| 1 | 9 (27.3%) | 13 (38.2%) |
| >1 | 7 (21.2%) | 8 (23.5%) |
| Duration of depression (weeks), mean (SD) | 24.6 (16.9) | 45.1 (121.9) |
| **Depressive episodes in past** | | |
| 0 | 12 (36.4%) | 11 (32.4%) |
| 1 | 9 (27.2%) | 14 (41.2%) |
| >1 | 12 (36.4%) | 9 (26.5%) |
| **Chronotype** | | |
| Early (extremely, moderately and slightly) | 20 (80%) | 25 (92.6%) |
| Normal | 1 (4%) | 1 (3.7%) |
| Late (extremely, moderately and slightly) | 4 (16%) | 1 (3.7%) |

BLT, bright light therapy; BMI, body mass index; DRLT, dim red light therapy.

women treated with BLT, eight women (24.2%) never missed a treatment, in contrast to three women (8.8%) in the DRLT group. Sixteen women (48.5%) treated with BLT missed a maximum of six treatments, compared with 20 women (58.9%) in the DRLT group. In both groups, two women missed 7–13 treatments in the intervention period. One woman treated with BLT and two with DRLT missed 14 or more treatments. One woman treated with BLT and two with DRLT missed the final 2 weeks of treatment, the first one due to complete remission of her symptoms.

## Maintaining blinding

Before treatment, three women (4.8%) did not expect any effect from light therapy for their depressive symptoms. All other participants expected a (small) positive effect. After treatment, one participant treated with BLT (3.0%) and three women in the group treated with DRLT (8.8%) thought they were treated with placebo treatment. All other women had no specific ideas about their allocation.

## Treatment effect

Online supplemental table 2 shows the observed median SIGH-SAD, HAM-D and EPDS Scores over the course of the study. In the women treated with BLT, median depression scores decreased by 42.6% (SIGH-SAD), 53.1% (HAM-D) and 40.6% (EPDS) in the intervention period. In the DRLT group, this was respectively 50.9%, 66.7% and 59.4%. After women stopped with light treatment, median scores continued to decrease for all questionnaires in both groups, 3 and 10 weeks after treatment. At 2 months postpartum, women treated with BLT showed no increase in EPDS Scores, whereas women treated with DRLT showed an increase in EPDS Scores. For both SIGH-SAD and HAM-D Scores, a decrease was observed in both treatment arms.

We also calculated the median improvement scores without the baseline score. For women treated with BLT, these were 6.1% (SIGH-SAD), 16.7% (HAM-D) and 13.6% (EPDS). For women treated with DRLT, this was, respectively, 31.6%, 40% and 45.8%.

No statistically significant difference was found between the two treatment arms for the intervention period, nor for the entire study. For the SIGH-SAD, our primary endpoint, we found β=−0.68 (95% CI −1.84, 0.49) for the intervention period and β=−0.16 (95% CI −0.82, 0.51) for the entire study (figure 2 and table 3). Adjusted primary analyses, where we repeated our primary analyses adjusted for propensity scores, and sensitivity analyses with imputed data did not show any other findings (online supplemental table 3). Adjustment for chronotype and month of treatment did not change our findings as well. Post-hoc analyses, where we repeated the analyses for women with higher treatment compliance and for women with higher symptom severity at baseline, did not show a statistically significant difference between the two treatment arms (online supplemental table 3).

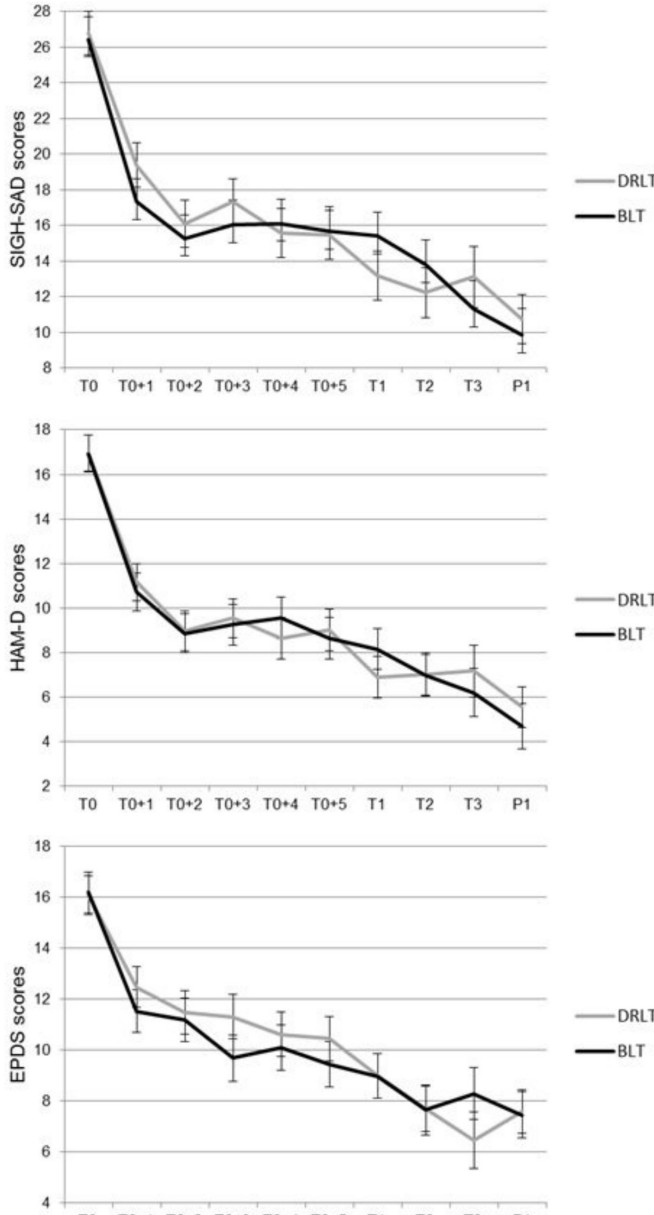

**Figure 2** Estimated marginal means of depression scores in women with antepartum depression until 2 months postpartum. Shown are SIGH-SAD, HAM-D and EPDS Scores. Black lines represent treatment with BLT, grey lines with DRLT. Bars represent SE of the mean. BLT, bright light therapy; DRLT, dim red light therapy; EPDS, Edinburgh Postnatal Depression Scale; HAM-D, Hamilton Rating Scale for Depression; SIGH-SAD, Structured Interview Guide for the Hamilton Depression Scale—Seasonal Affective Disorder version. T0, baseline, before treatment; T0+1, T0+2 … T0+5, weeks during intervention period; T1, end of treatment; T2, 3 weeks after end of treatment; T3, 10 weeks after end of treatment; P1, 2 months postpartum.

For the HAM-D, 13 participants in the BLT group and 17 participants in the DRLT group were considered responders. This was, respectively, 11 and 9 when measured with the EPDS. When we studied responders versus non-responders, we found no statistically significant

**Table 3** Effects of allocation on the course of depressive symptoms through the intervention period and follow-up (until 2 months postpartum): crude analysis

|  | β (95% CI) of intervention* | β (95% CI) of follow-up† |
|---|---|---|
| SIGH-SAD | −0.68 (−1.84, 0.49) | −0.16 (−0.82, 0.51) |
| HAM-D | −0.18 (−0.74, 0.37) | 0.04 (−0.29, 0.37) |
| EPDS | 0.01 (−0.51, 0.53) | −0.05 (−0.35, 0.24) |

*From start of study until end of treatment.
†From start of study until follow-up 2 months postpartum.
EPDS, Edinburgh Postnatal Depression Scale; HAM-D, Hamilton Rating Scale for Depression; SIGH-SAD, Structured Interview Guide for the Hamilton Depression Scale—Seasonal Affective Disorder version.

differences for both HAM-D Scores (p=0.46) and EPDS Scores (p=0.60).

## Side effects

For women treated with BLT, the most frequently reported side effect was headaches (30.3%), followed by sleep problems (12.1%) and nausea (6.1%). For women treated with DRLT, the most reported side effect was headaches (20.6%), followed by sleep problems (8.9%) and irritable eyes (5.9%). Side effects were not reported more often by women treated with BLT, compared with DRLT (p=0.52). Most side effects were experienced for a maximum of 3 days. None of the women suffered from any (hypo)manic symptoms. We reduced the treatment duration for five women to 20 min daily due to their side effects. Interestingly, two women dropped out of the study due to side effects, but only in the DRLT group.

## Acceptability and satisfaction

The majority of women experienced a (small) positive effect for their depressive symptoms (78.6% BLT; 61.5% DRLT; p=0.58). All participants found the lamp (very) easy in use. Most women found the light therapy pleasant (57.1% BLT; 50% DRLT; p=0.49). Twenty-six women reported that it was (very) easy to plan the light therapy in the morning (42.9% BLT; 53.8% DRLT; p=0.43). Thirty-two women reported that they would like to use light therapy outside of the study (57.1% BLT; 61.5% DRLT; p=0.79). On average, women reported it was likely they would recommend the light therapy to others (BLT mean 8.0, SD 1.3; DRLT mean 7.0, SD 2.7; p=0.08).

## DISCUSSION

We conducted an RCT, evaluating the effectiveness of BLT in a sample of 67 pregnant women with major depressive disorder, compared with DRLT. We found no statistically significant difference between BLT and DRLT on depressive symptoms. Median depression scores decreased by 40.6%–53.1% during the intervention in the women treated with BLT and by 50.9%–66.7% in the women treated by DRLT.

### Effects in the current study

This level of improvement is comparable to the studies by Oren et al[51] and Corral et al[71] who both found a reduction in mean depression scores of 49%. Oren et al conducted an open trial in an antepartum population, whereas Corral et al conducted an RCT among women with a postpartum depression. Similar to Corral et al, we did not find a statistically significant difference between the effective and placebo conditions. The median improvement in the DRLT group can be explained by placebo effects, which could also be the case in the BLT group. A meta-analysis showed that the placebo response in antidepressant trials is approximately 68%,[72] although this effect is not clear yet in light therapy trials specifically. Second, the improvement in both groups can be explained by non-specific treatment effects such the structure offered by the study,[43] the interaction with the researchers or increased awareness and self-care resulting from participating in the study. A systematic review on various studies in treating antepartum depression with a control condition showed that these trials often show a considerable reduction in symptom scores in both treatment arms.[39] Furthermore, it might be that symptoms decrease related to the course of pregnancy, spontaneous remission or regression to the mean. A meta-analysis showed that untreated depressive symptoms could decrease by 10%–15%, on average.[73] However, untreated depression during pregnancy is an important predictor for postpartum depression.[74] We calculated the improvement of the depressive symptoms without the baseline scores, to study whether the improvement was especially notable in the first week of treatment. We found that the improvement was less, especially in the group treated with BLT, which may pinpoint to regression to the mean. For example, women may have the feeling of 'finally being heard', or feeling empowered about doing something about their symptoms, which may explain these findings.

Corral et al mentioned that several participants commented positively on having 30 min of 'quiet time' on a daily basis. Several of our participants mentioned this as well, which could reflect sinking into a state of more relaxation or more mindfulness which may have contributed to the improvement in both groups. Two meta-analyses showed that mindfulness-based therapy is an effective treatment for a variety of psychological problems.[75 76] An earlier pilot study and an open study of mindfulness also showed positive effects on mood specifically in pregnant women.[77 78] Corral et al mentioned that many postpartum women are motivated to access recourses, such as psychological treatment, which could have exerted non-specific treatment effects. In their study, however, no participant took part in any treatment during the study. In our study, several women started psychotherapy or antidepressant medication. However, adjustment for any intervention did, however, not change our findings.

Finally, it has been shown earlier in healthy volunteers that treatment with similar conditions as our placebo therapy might actually have some effects in melatonin

suppression,[79] which could explain why we actually see a decrease of symptoms in the DRLT group.

## Differences with literature

The results of this study differ from the RCTs by Epperson et al[52] and Wirz-Justice et al,[53] who did find superiority of BLT over placebo in an antepartum population.

Wirz-Justice et al included only clinical patients and found that BLT had more effects in severe patients in their study. However, mean baseline SIGH-SAD Scores in the Wirz-Justice et al and Epperson et al studies were 27.7 and 28.1, respectively, which were not clinically relevant different from the present study (26.5). Additionally, we included baseline depression scores in our model, which did not change our findings. Also, post-hoc analyses, where we repeated the analyses for women with higher baseline severity, did not show any significant findings.

Both Epperson et al and Wirz-Justice et al treated their patients for 1 hour a day and within 10 min of habitual wake-up time, which is different from the present study. Thus far, no studies have been executed comparing the effectiveness of shorter versus longer exposure to bright light in non-seasonal depression. Possibly, more light output in the BLT group would be necessary to show superiority of BLT over DRLT in a pregnant population. However, other studies that treated patients for 30 min also did show a statistical significant difference between the effective and the placebo intervention in non-seasonal depression.[46] One must keep in mind that these studies have been done in non-pregnant populations and different—yet unknown—underlying mechanisms may play a part during pregnancy, such as hormonal fluctuations and a shift in social role.

Our placebo condition, in which the possible effect of DRLT could be questioned, is not a plausible explanation for not finding a statistically significant effect between the treatment arms. Epperson et al used a placebo condition with 500 lux white light, which is questionable as a placebo, for white light of 100 lux is able to phase-shift human circadian rhythms.[80] Since this study found a significant improvement in women treated with BLT when compared with this placebo, it is unlikely that the settings of our placebo would explain failing to achieve a significant difference between the two treatment arms.

In the study by Corral et al, depression scores worsened after withdrawal of treatment, indicating that spontaneous remission would be less likely. However, in the present study, median depressions scores of all questionnaires continued to improve after withdrawal of treatment in both groups, indicating that spontaneous remission in both groups is a possible explanation for this finding.

## Strengths and limitations

Internationally, we conducted the largest RCT studying light therapy in pregnant women with a depression. Moreover, we conducted various follow-up measurements, including postpartum, to study the effects of withdrawal of treatment and to study whether treatment during pregnancy would protect against postpartum depression. Another strength is using a single assessor to diagnose depression. Moreover, the setting of treatment was within a real-world setting. Finally, a strength of this study was the comprehensive assessment of side effects, as well as acceptability and satisfaction of treatment.

The main limitation of our study was that an unforeseen lack of resources prevented us from including 150 participants, as we aimed to do according to our sample size calculation,[55] which enables us to find only large treatment effects.[55] Another limitation is the fact that depressive symptoms during the study are assessed by questionnaires, rather than diagnostic criteria. Also, information about psychiatric history was collected via an interview and not through medical records, which may be influenced by recall bias. Moreover, various covariates are self-reported, such as BMI, substance use and medication. We noticed a different attrition rate at T3 (10 weeks after treatment) and P1 (2 months postpartum). At T3, this is due to the fact that more women treated with DRLT already gave birth at T3, which resulted in missing data. We do not have an explanation for the different attrition rate at P1. We cannot rule out the possibility that these differences in attrition might have impacted our follow-up results. However, our sensitivity analyses indicate our follow-up results to be robust for differences between the conditions and data imputation.

## Conclusions

BLT has been shown effective in treating non-seasonal depression[46] and in women with antepartum depression as well.[52 53] In the present study, depressive symptoms of pregnant women with depression improved in both treatment arms after 6 weeks of treatment. Given the very mild and short-lived side effects, the major improvement in a short time period, the high acceptability of the participants, the low costs and the direct availability, more studies to the effectiveness of BLT during pregnancy are warranted. It is important to determine whether the responses observed in the present study represent true treatment effects, non-specific treatment responses, placebo effects or a combination of these. This could be done by studying biological outcomes, such as cortisol and melatonin levels, which might show a statistically significant difference between the two treatment arms irrespective of perceived symptoms of depression. Additionally, it might show an indication of the positive effects of light therapy on the circadian rhythm and its inhibiting effects on HPA-axis hyperactivity.

**Acknowledgements** We would like to thank all participants for participating in the study. We would also like to thank all general practitioners, midwifes, gynaecologists, psychiatrists and psychologists for their help with the recruitment. We are grateful for all coworkers, students and assistants who contributed to the data collection in this study: Nina Molenaar, PhD, Marlies Brouwer, PhD, Leo Genet, MSc, Sophie de Droog, MSc, Sofie Koomen, MSc, Diewertje Houtman, MSc, Maria Zepeda, MSc, Nicolle Croes, MSc, Rianne Winters, MSc. Lisanne van Kesteren, BSc, Finn Stofkoper, BSc, Indira Schouten, MSc and Mieke Roukema, MSc.

**Contributors** ML-vdB is the project's principle investigator and initiator of the study, obtained funding and designed the study. BB was responsible for recruiting and counselling participants, running the study and collecting data. AMK is the project's methodologist and executed the primary statistical analysis. HHB and EK were involved in the recruitment of the study. JLS provided support. AMK, WJGH and ML-vdB supervised the study. BB, AMK and ML-vbB prepared the original draft. All authors reviewed, edited and approved the final manuscript.

**Funding** MLB received funding from the 'Light, Cognition, Behaviour and Health' program of The Netherlands Organization for Health Research and Development (NWO; The Hague, The Netherlands), in collaboration with Signify Research (grant number 058-14-003) to fund the current study.

**Competing interests** JLS is employed by Signify Research. The lamps used in this study were provided by Signify Research.

**Patient consent for publication** Not required.

**Ethics approval** All procedures performed involving human participants were in accordance with the ethical standards of the institutional and/or national research committee and with the 1964 Helsinki declaration and its later amendments or comparable ethical standards. Written informed consent was obtained from all participants. The study protocol and later amendments were approved by the medical ethical committee of the Erasmus University Medical Centre, Rotterdam, The Netherlands (registration number MEC-2015–731).

**Provenance and peer review** Not commissioned; externally peer reviewed.

**Data availability statement** Data are available upon reasonable request. The datasets used and/or analysed during the current study are available from the author Mijke Lambregste-van den Berg on reasonable request.

**ORCID iD**
Babette Bais http://orcid.org/0000-0003-3579-6670

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
