## [Reviewer comments · BMJ Open]

ARTICLE DETAILS

TITLE (PROVISIONAL)	Effects of bright light therapy for depression during pregnancy: a randomized, double-blind controlled trial
AUTHORS	Bais, Babette; Kamperman, Astrid M.; Bijma, Hilmar; Hoogendijk, Witte; Souman, Jan; Knijff, Esther; Lambregtse-van den Berg, Mijke

VERSION 1 – REVIEW

REVIEWER	Morgan R. Peltier NYU-Winthrop Hospital NYU-Long Island School of Medicine USA
REVIEW RETURNED	27-Dec-2019

GENERAL COMMENTS	This is a very good paper for which I have only a few comments. 1. I don't think this quite qualifies as a double blind study because the therapy cannot be hidden from the patients and there is some knowledge out there about an association between depression and light. Even if the patient's are lied to/misled about the nature of their treatment, it's likely they'll know which group they are in and which may be more or less effective. Also knowing that they are receiving a phototherapy of some type would may have caused them to feel like they "should be better". Although this would not be a differential bias, it could be responsible for the Hawthorne-type effect that was observed where everyone reported improvements.2. Generalized mixed effects models are appropriate for these studies but the authors should probably used the extensions that allow for analysis of ordinal or Poisson-distributed data since scores are non-continuous outcomes. Results should be presented as medians and ranges. Slopes in table 3 can be presented as they are or exponentiated and presented as relative risks. That said, I don't think this more appropriate analysis it would have affected their findings. No difference between DRLT and BLT was detected with Gaussian methods and both ordinal and Poisson-regression methods would have less statistical power. Minor suggestions: Introduction: Please also list inflammation as a possible cause of antenatal depression and discuss if there are variations in rates of diagnosis of antenatal depression with season or geographic latitude.
---

	Table 1. Please add a column of P-values for the comparisons (with Chi², independent t-tests, fisher's exact test, or non-parametric equivalents as appropriate) between BLT and DRLT patients. If there are no statistically significant (or even with only marginally significant differences) differences between groups for all these demographic and background factors, there is no need to do the adjusted analysis with propensity scores. Also indicate in this part of the results section that there were no differences between groups in these factors. I believe that this is key to demonstrating the success of your randomization. Table 3, Please use a comma to separate lower and upper 95% CI.
--	--

REVIEWER	Teresa Neeman Australian National University, Australia
REVIEW RETURNED	02-Jan-2020

GENERAL COMMENTS	Major concerns: The authors report a very large drop in depression scores (SIGH-SAD) in the abstract (~50%), but it was clear from Supp Table 2 that this drop was driven by a large change between baseline and Week 1 scores. The authors offer no explanation for this huge drop. It is important to address this because that single drop drives the reported estimates for improvement. In the absence of any other information, one looks for reasons for this unusual pattern. Here are two possibilities: (1) The baseline test was done over the telephone, whereas all subsequent tests were done online. It is well-known that the mode of assessment can have a large impact on responses. (2) Regression to the mean is another wellknown phenomenon in RCTs. When patients are assessed exactly once, and selected for high scores, then the baseline scores the selected patients will tend to be higher than their "true" normal score, and subsequent scores will be lower, making the treatment look like it is beneficial. There is evidence of this selection 'bias': 13 subjects were excluded for insufficiently high scores. For these two reasons, the baseline scores are "problematic". What happens if one fits the random slopes model (Time x treatment) to the data, after removing the baseline score? How does this change the estimates of improvement in each group? If it changes it a lot, how comfortable do the authors feel about reporting a 50% improvement in symptoms, given that the baseline value is unusually high? Other comments: Table 3 gives the main results from the proposed primary analysis, but these are not reported in the abstract. Conversely, the main results reported in the abstract are not the results from the proposed primary analysis (linear mixed model with time x treatment as fixed effects). Table 3 reports (I think) the difference in the slope estimates between interventions. It would be clearer to report the estimated slopes (and 95% CI) for each group separately. These are the estimates of most interest to the clinician. The difference between the two
---

	slopes is important for inference, but not so meaningful on their own. Table 3: It's unclear how beta was calculated in the second column (follow up). The title suggests that it refers to the change across the follow-up period, but the caption suggests it is change estimate across the entire study. Table 3: I recommend that you put the estimates for the sensitivity analyses in a supplementary table (just to demonstrate that you did them, and the additional analyses didn't change your conclusions). I had a look at the sample size estimate in the protocol. I wasn't able to reproduce this estimate because of missing information about the within-group variation. But even with that information, I think I would be unable to reconstruct the sample size estimation, because there was no clearly defined model. Missing data: I think there could have been some discussion around loss to follow-up. This reader assumed that dropout was due to higher depression scores (ie informative censoring), and it would be helpful if the authors thought about this, and discussed how this might bias their analysis. The paper refers to 283 enrolments, but this is misleading. They should instead use the word "referrals".
--	--

REVIEWER	Noha S. Daher Loma Linda University School of Allied Health Professions Loma Linda, CA United States
REVIEW RETURNED	08-Jan-2020

GENERAL COMMENTS	The authors analyzed the data and displayed the results properly. However, I have few comments regarding the study.  1. There are many grammatical errors and the authors move from present to past tense . Line 74 "have" instead of "has"; line 87 "influence" not "influencing",... 2. At the end of the introduction, the purpose needs to be addressed using past tense since the study was done and needs to be more clear "We compared the effectiveness of BLT therapy compared to placebo light among pregnant women with a depressive disorder" In addition, we followed....hypothesized.... 3. Under "Participants", line 106, this is the research design and not describing participants....The purpose needs to be removed from this section. Start with "Eligible..." 4. Lines 165-167 need to be moved to the discussion section 5. Line 204, questionnaires were given to the participants. What is body material? 6. Line 257, summarized using mean...; line 258, summarized by count and percent. Line 261, we included all observations until the study ended, or the participant(s) dropped out of the study. 7. Under results, lines 292-293, Mean (SD) of SiGH....and delete all the other repetitions of mean and SD. Line 310, delete "where this was". Change it to compared to... 8. In the "Discussion section", authors need to comment on the attrition rate and its effect on the internal validity of the study. In Supplementary Table 2, at T3 and P1 we have different attrition rates for BLT and DRLT. For this Table, all abbreviations need to be explained below it. Same with the graph. For Table 2, Change the title to Frequency (%) of participant characteristics by study
--

	group... and remove % from the body of the table, and explain the abbreviations (BLT and DRLT in a footnote.....
--	--

REVIEWER	Krzysztof Krysta Medical University of Silesia, Katowice, Poland
REVIEW RETURNED	12-Jan-2020

GENERAL COMMENTS	This is a very important paper on the use of bright light therapy in depressive pregnant women. According to numerous data BLT may very useful in this group of patients. In general the paper has a good study design and methodology. The results look very interesting and they are supported by a good discussion. However I have a few detailed questions:  - In the Methodology it was stated that "Previous depressive episodes were also assessed with the SCID." Could the Authors explain more precise how it was done - using just oral memories of the patients, medical documentation, etc. It was mention that psychiatric history was collected via a telephone interview Collecting data of previous depressive episodes may often bring very subjective data. - Did previous pharmacological treatments (e.g. treatment resistant depression) impact the inclusion process? Did the Authors exclude patients with previous ECT treatment? - It was said that: During the entire study, four additional women started with psychotropic medication". Was it the first time in their life the used antidepressants? If not did they use the same medication as earlier?
--

VERSION 2 – REVIEW

REVIEWER	Krzysztof Krysta Department of Rehabilitation Psychiatry Medical University of Silesia, Katowice, Poland
REVIEW RETURNED	17-Mar-2020

GENERAL COMMENTS	I would like to thank the Authors for all changes and improvements they have done. I have no more comments.
---

REVIEWER	Yael Nillni Boston University School of Medicine USA
REVIEW RETURNED	21-May-2020

GENERAL COMMENTS	This study examined the effects of bright light therapy on depression during pregnancy. The authors describe the results of a well-designed and conducted study and were responsive to reviewer comments on the initial submission of this manuscript. I have some minor remaining comments noted below. General 1. The manuscript would benefit from a careful read through to catch grammatical errors. For example In the participants section: "...number of women was referred..." should be "...number of women were referred..." in the results section: "The majority of the participants (82%) was recruited..." should be "The majority of the
---

participants (82%) were recruited...” In the discussion section: “However, mean baseline SIGH-SAD score in the Wirz-Justice et al. and Epperson et al. studies were 27.7 and 28.1, respectively, which is not clinically relevant different from the present study (26.5).” should be “However, mean baseline SIGH-SAD scores in the Wirz-Justice et al. and Epperson et al. studies were 27.7 and 28.1, respectively, which were not different from the present study (26.5).” This is not an exhaustive list, just several examples.

Abstract

1. I think the word “women” is missing from the line describing participants. Should read 67 pregnant (12-32 weeks gestational age) women with a DSM-5 diagnosis of depressive disorder.

Introduction

1. The authors added this sentence based on reviewer feedback: “Antepartum depression is not only seen in autumn and winter, but is a year-round phenomenon, with certain subgroups even showing more symptoms in summer.” This sentence is really out of place in the flow of the paragraph. The authors are talking about antepartum depression, which has nothing to do with seasonal depression. I also don’t think it address the reviewer’s comment. The reviewer asked the authors to speak to whether rates of antenatal depression differ by season or geographic latitude. If there are no data on rate differences to report, I would just simply reword this sentence to describe the prevalence of antepartum depression and that differences in antepartum depression by season and geographic latitude is unknown. Similarly, the addition of inflammation as a reason for antepartum is random and out of place here. If you are going to add information about mechanisms for antepartum depression, I would include more than just inflammation. It may be possible the reviewer meant for you to add inflammation to the list of mechanisms impacting HPA axis programming in the fetus?

2. This sentence is missing some descriptive information: “Additionally, children show more often cognitive, emotional and behavioral problems in childhood, adolescence and adulthood 10,11 and they have a higher risk of suffering from depression later in life.” Children of mothers with antepartum or postpartum depression?

Methods

1. It reads strange in a manuscript to bullet the follow-up time points. I would just list them in the paragraph.

2. The authors state that they “collected urine, hair and cortisol from the participants, as can be found in our earlier published protocol” Do you mean collected urine, hair, and blood samples? Or urine, hair, and saliva samples? I would just list what was actually collected. I don’t think you need to mention cortisol.

3. The authors state that the rationale for using SIGH-SAD as the primary outcome measure was because this is what is typically assessed in light therapy trials. However, this is not a seasonal affective disorder sample, so why not use the HAM-D or the EPDS as the primary outcome given that you are looking at nonseasonal depression?

Results

	1. The authors should include some language in text about any significant differences between groups in comorbidity, baseline depression scores, initiation of antidepressants or psychotherapy. The authors describe this for the whole sample, but not by group. Was nothing significantly different? If so, I would state that. For example, looking at table 2 it seems like more women in the BLTR group initiated antidepressants than the BLT group? Was that a significant difference? If so, could that be a reason for improvement in the BLTR group? Tables/Figures 1. I don't see any figures in the manuscript?
--	--

REVIEWER	Mike Bradburn University of Sheffield
REVIEW RETURNED	26-Jul-2020

GENERAL COMMENTS	I have four comments - I believe all are important but can be addressed. 1) The authors appeared to have addressed most of the previous reviewers comments. There are however two which I want to highlight (both reviewer 2) a) I was also unable to reproduce the sample size, even with the additional information supplied to reviewers comments. Most specifically, the sample size depends on the size of the difference in relation to the standard deviation (neither are defined). I assume the test statistic is based on the time*treatment term, but the number of timepoints is not stated. Furthermore this interaction could reasonably use time as a linear term (which looks for a gradual separation) or as a discrete time (which would measure the area between the curve). b) On the other hand I disagree with the reviewer on the importance of within-group changes. The point of a controlled trial is to have a non-treated group to compare against. The change within each group is informative and I am happy for it to be retained, but the primary interest is the difference in effectiveness between the two groups. Please add a measure of difference together with its confidence interval (CI) to the results. This is particularly important given the under-recruitment and the resultant imprecision in the difference between arms: a CI will help give an idea how large any difference is likely to be. 2) Abstract - the first sentence says both arms were effective: this is not quite right. It is true that both arms improved, but "effective" implies this was caused by the trial treatment. The change may have happened for other reasons, as noted in the second sentence (regression to the mean is another possible explanation) 3) Methods - the subtitle "Confounders" is probably not appropriate? Confounders are things which cannot be separated from a treatment effect, which should be balanced by randomisation. Perhaps "baseline characteristics" ? 4) Statistical methods - as above please specify whether time was considered categorical or continuous.
--

VERSION 2 – AUTHOR RESPONSE

Reviewer: 1

I would like to thank the Authors for all changes and improvements they have done. I have no more comments.

We would like to thank reviewer 1 for his/her comments on the manuscript, thereby providing a valuable contribution to improving the manuscript.

Reviewer: 2

This study examined the effects of bright light therapy on depression during pregnancy. The authors describe the results of a well-designed and conducted study and were responsive to reviewer comments on the initial submission of this manuscript. I have some minor remaining comments noted below.

First of all, we would like to thank reviewer 2 for his/her valuable comments on our manuscript.

General

1. The manuscript would benefit from a careful read through to catch grammatical errors. For example In the participants section: “....number of women was referred...” should be “...number of women were referred...” in the results section: “The majority of the participants (82%) was recruited...” should be “The majority of the participants (82%) were recruited...” In the discussion section: “However, mean baseline SIGH-SAD score in the Wirz-Justice et al. and Epperson et al. studies were 27.7 and 28.1, respectively, which is not clinically relevant different from the present study (26.5).” should be “However, mean baseline SIGH-SAD scores in the Wirz-Justice et al. and Epperson et al. studies were 27.7 and 28.1, respectively, which were not different from the present study (26.5).” This is not an exhaustive list, just several examples.

We apologize for these grammatical errors, as it is custom to write it this way in Dutch. We went through the manuscript and corrected these and other sentences accordingly.

Abstract

1. I think the word “women” is missing from the line describing participants. Should read 67 pregnant (12-32 weeks gestational age) women with a DSM-5 diagnosis of depressive disorder.

Thank you for pointing this out to us, it was indeed missing. We added the word “women” to line 25 of the manuscript.

Introduction

1. The authors added this sentence based on reviewer feedback: “Antepartum depression is not only seen in autumn and winter, but is a year-round phenomenon, with certain subgroups even showing

more symptoms in summer.” This sentence is really out of place in the flow of the paragraph. The authors are talking about antepartum depression, which has nothing to do with seasonal depression. I also don’t think it address the reviewer’s comment. The reviewer asked the authors to speak to whether rates of antenatal depression differ by season or geographic latitude. If there are no data on rate differences to report, I would just simply reword this sentence to describe the prevalence of antepartum depression and that differences in antepartum depression by season and geographic latitude is unknown. Similarly, the addition of inflammation as a reason for antepartum is random and out of place here. If you are going to add information about mechanisms for antepartum depression, I would include more than just inflammation. It may be possible the reviewer meant for you to add inflammation to the list of mechanisms impacting HPA axis programming in the fetus?

Thank you for your valuable comment. Indeed, the reviewer asked whether rates of antepartum depression differ by season or geographic latitude. The study we refer to [5] is actually a study we have done in the past. Here, we studied whether rates of antepartum depression differed through the year. We found that antepartum depression (using a cut-off of EPDS ≥ 9) was found all year through with a prevalence of 13.2%. We found a distinct difference between two groups, where one group showed a pattern as one would expect, with more symptoms in winter and less in summer. The other group showed a pattern opposite from this, with more symptoms in summer and less in winter. However, this was not a study in women suffering from seasonal depression. This was a large-scale cross-sectional study in screening women for psychopathology, psychosocial problems, and substance abuse, not necessarily women only suffering from antepartum or seasonal depression. Therefore, we do think that it addresses the comment of the reviewer. To our knowledge, no studies have been executed so far studying rates of antepartum depression by geographical latitude.

Regarding the comment about inflammation: we agree that the addition of inflammation may be random here. We have changed the sentence to the following (line 61-62): “Possible causes for antepartum depression may include alterations in endocrine systems, such as the hypothalamus-pituitary-adrenal axis, and inflammation.”

2. This sentence is missing some descriptive information: “Additionally, children show more often cognitive, emotional and behavioral problems in childhood, adolescence and adulthood and they have a higher risk of suffering from depression later in life.” Children of mothers with antepartum or postpartum depression?

Thank you for pointing this out to us. We have added “of mothers with antepartum depression” (line 66).

Methods

1. It reads strange in a manuscript to bullet the follow-up time points. I would just list them in the paragraph.

We changed follow-up time points to listing it in the paragraph.

2. The authors state that they “collected urine, hair and cortisol from the participants, as can be found in our earlier published protocol” Do you mean collected urine, hair, and blood samples? Or urine, hair, and saliva samples? I would just list what was actually collected. I don’t think you need to mention cortisol.

Thank you for noticing this. Indeed, we did not mean cortisol, we meant saliva. We have changed it accordingly (line 219).

3. The authors state that the rationale for using SIGH-SAD as the primary outcome measure was because this is what is typically assessed in light therapy trials. However, this is not a seasonal affective disorder sample, so why not use the HAM-D or the EPDS as the primary outcome given that you are looking at nonseasonal depression?

This is indeed not a trial studying seasonal affective disorder, so we understand the confusion. However, since the current study was a light therapy trial, which most often uses the SIGH-SAD as a primary outcome measure, we decided to use this questionnaire as well, in order to be more comparable to earlier conducted studies. Moreover, our power calculation was based on effect sizes which were found with the SIGH-SAD.

Results

1. The authors should include some language in text about any significant differences between groups in comorbidity, baseline depression scores, initiation of antidepressants or psychotherapy. The authors describe this for the whole sample, but not by group. Was nothing significantly different? If so, I would state that. For example, looking at table 2 it seems like more women in the BLTR group initiated antidepressants than the BLT group? Was that a significant difference? If so, could that be a reason for improvement in the BLTR group?

In our sensitivity analyses, we adjusted for various patient characteristics, such as antidepressant use and psychotherapy. Our analyses did not show any differences between the two groups. Also, in our post-hoc analyses, we tested whether women with more severe depressive symptoms at baseline responded differently. Again, these analyses did not show any statistical significant differences (see lines 349-355). Therefore, we concluded that these characteristics did not impact the improvement in both groups.

Tables/Figures

1. I don't see any figures in the manuscript?

In the previous PDF proofs, the figures can be found at the end of manuscript (page 29 and 30).

Reviewer: 3

I have four comments - I believe all are important but can be addressed.

We would like to thank reviewer 3 for his/her comments.

1) The authors appeared to have addressed most of the previous reviewers comments. There are however two which I want to highlight (both reviewer 2)

a) I was also unable to reproduce the sample size, even with the additional information supplied to reviewers comments. Most specifically, the sample size depends on the size of the difference in relation to the standard deviation (neither are defined). I assume the test statistic is based on the time*treatment term, but the number of timepoints is not stated. Furthermore this interaction could reasonably use time as a linear term (which looks for a gradual separation) or as a discrete time (which would measure the area between the curve).

Before the start of the study, sample size was calculated using GLIMMPSE 2.1.5. software, with the following parameters: alpha 0.05; beta 0.80; 6 time assessments (continuous, equally spaced); primary test: time*treatment interaction; SIGH-SAD scores assumed at baseline: M:28.0 and SD:7.0; Hotelling-Lawley Trace correction; base correlation 0.4; decay rate 0.05; no additional scaling factors included. Power analyses were conducted for a range of effect sizes from small 10% to medium 15% change difference.

b) On the other hand I disagree with the reviewer on the importance of within-group changes.

The point of a controlled trial is to have a non-treated group to compare against. The change within each group is informative and I am happy for it to be retained, but the primary interest is the difference in effectiveness between the two groups. Please add a measure of difference together with its confidence interval (CI) to the results.

This is particularly important given the under-recruitment and the resultant imprecision in the difference between arms: a CI will help give an idea how large any difference is likely to be.

In Supplementary Table 3, we have shown the effects of allocation on the course of the symptoms, both for the intervention period as for the entire study. Here, we have also shown the confidence intervals. We assume this is what the reviewer is referring to. Since we have no paired observations between the allocations, it was not possible to calculate mean difference scores between both treatment conditions.

2) Abstract - the first sentence says both arms were effective: this is not quite right. It is true that both arms improved, but "effective" implies this was caused by the trial treatment. The change may have happened for other reasons, as noted in the second sentence (regression to the mean is another possible explanation)

We have changed the sentence to the following: "BLT and DRLT both reduced depressive symptoms in pregnant women with depression."

3) Methods - the subtitle "Confounders" is probably not appropriate? Confounders are things which cannot be separated from a treatment effect, which should be balanced by randomisation. Perhaps "baseline characteristics" ?

We agree with reviewer 3 and changed the subtitle to "Baseline characteristics".

4) Statistical methods - as above please specify whether time was considered categorical or continuous.

Mixed model analyses are conducted using time as a continuous factor. We added this in the statistical analyses (line 282).

References

1. Epperson, C.N., et al., Randomized clinical trial of bright light therapy for antepartum depression: preliminary findings. *J Clin Psychiatry*, 2004. 65(3): p. 421-5.
2. Wirz-Justice, A., et al., A randomized, double-blind, placebo-controlled study of light therapy for antepartum depression. *J Clin Psychiatry*, 2011. 72(7): p. 986-93.
3. Bais, B., et al., Bright light therapy in pregnant women with major depressive disorder: study protocol for a randomized, double-blind, controlled clinical trial. *BMC Psychiatry*, 2016. 16(1): p. 381.
4. Williams JBW, L.M., Rosenthal NE, Amira L, Terman M, Structured Interview Guide for the Hamilton Depression Rating Scale, Seasonal Affective Disorders version (SIGH-SAD). 1988, New York: New York Psychiatric Institute.
5. Bais, B., et al., Seasonality of depressive symptoms during pregnancy. *Psychiatry Res*, 2018. 268: p. 257-262.

VERSION 3 – REVIEW

REVIEWER	Yael Nillni National Center for PTSD, Women's Health Sciences Division at VA Boston Healthcare System and Boston University School of Medicine, USA
REVIEW RETURNED	21-Sep-2020

GENERAL COMMENTS	The authors have addressed all remaining comments.
--

REVIEWER	Mike Bradburn University of Sheffield, UK
REVIEW RETURNED	31-Aug-2020

GENERAL COMMENTS	I am concerned at two of the responses given to the previous review. This manuscript can be made publishable with very little modification, but there are two important misunderstandings in the rebuttal. This may be a language issue (I understand this is not the authors first lanhuage), and perhaps I haven't explained well enough. I hope the below is clearer. 1) "In Supplementary Table 3, we have shown the effects of allocation on the course of the symptoms, both for the intervention period as for the entire study. Here, we have also shown the confidence intervals. We assume this is what the reviewer is referring to. Since we have no paired observations between the allocations, it was not possible to calculate mean difference scores between both treatment conditions." -Yes, I requested (and more importantly, the CONSORT guidelines mandates) the text includes the difference in means together with a confidence interval, for the primary endpoint at a minimum. It is sufficient to report the differences in table 3 and to
---

	note that sensitivity analyses (cross-reference table S3) gave similar estimates and the same conclusion. Please add this alongside the change in each group. -I don't understand the comment "Since we have no paired observations between the allocations, it was not possible to calculate mean difference scores between both treatment conditions.". A difference in the means does not need to be calculated on paired data; rather, it needs to reflect the difference in means between the two groups over a period of time. I assume and hope that this is what supplementary table 3 reports. -Lastly (and I didn't notice this first time so I apologise), the results for SIGH-SAD, HAMD and EPDS in "adjusted analyses" are all identical for intervention and long term follow up. This is possible but unlikely - can you please check and confirm? 2) "We have changed the sentence to the following: "BLT and DRLT both reduced depressive symptoms in pregnant women with depression." -again I must disagree. It is certainly possible that both therapies work, but the the authors acknowledge (and reference) in the discussion that the change may be due to several other causes ("the improvement in both groups can be explained by non-specific treatment effects such the structure offered by the study[....] symptoms decrease related to the course of pregnancy, spontaneous remission, or regression to the mean"). It is very fair to say that symptoms reduced in both groups, but please avoid describing this causal - we simply don't know if it is this or other things (or a mixture). 3) Lastly, the details of the sample size need to be included in the text as well as in the authors response. I am still unable to reproduce the sample size using GLIMMIX (which appears to ask for the mean at each time point), although using a similar method in other software suggests that 63 per arm is about right.
--	--

VERSION 3 – AUTHOR RESPONSE

Reviewers' Comments to Author:

Reviewer: 2

The authors have addressed all remaining comments.

We would like to thank again reviewer 2 for his/her comments on the manuscript, thereby providing a valuable contribution to improving the manuscript.

Reviewer: 3

First of all, we would like to thank reviewer 3 for his/her time and effort in improving our manuscript.

I am concerned at two of the responses given to the previous review. This manuscript can be made publishable with very little modification, but there are two important misunderstandings in the rebuttal. This may be a language issue (I understand this is not the authors first language), and perhaps I haven't explained well enough. I hope the below is clearer.

Yes, English is indeed not our first language. We appreciate the time you take to explain things to us.

1) "In Supplementary Table 3, we have shown the effects of allocation on the course of the symptoms, both for the intervention period as for the entire study. Here, we have also shown the confidence intervals. We assume this is what the reviewer is referring to. Since we have no paired observations between the allocations, it was not possible to calculate mean difference scores between both treatment conditions."

-Yes, I requested (and more importantly, the CONSORT guidelines mandates) the text includes the difference in means together with a confidence interval, for the primary endpoint at a minimum. It is sufficient to report the differences in table 3 and to note that sensitivity analyses (cross-reference table S3) gave similar estimates and the same conclusion. Please add this alongside the change in each group.

Thank you for this suggestion. We have added the following to the results (line 353-354 in the manuscript with track changes): "For the SIGH-SAD, our primary endpoint, we found $\beta = -0.68$ (95% CI -1.84, 0.49) for the intervention period and $\beta = -0.16$ (95% CI -0.82, 0.51) for the entire study." We decided to only report our primary endpoint, for we would otherwise present 6 different numbers including confidence intervals, which would make the text less readable.

Lines 355-357 already included the following: "Adjusted primary analyses, where we repeated our primary analyses adjusted for propensity scores, and sensitivity analyses with imputed data did not show any other findings (Supplementary Table 3)."

-I don't understand the comment "Since we have no paired observations between the allocations, it was not possible to calculate mean difference scores between both treatment conditions.". A difference in the means does not need to be calculated on paired data; rather, it needs to reflect the difference in means between the two groups over a period of time. I assume and hope that this is what supplementary table 3 reports.

Yes, this is what Supplementary Table 3 reports, we apologize for the misunderstanding.

-Lastly (and I didn't notice this first time so I apologise), the results for SIGH-SAD, HAMD and EPDS in "adjusted analyses" are all identical for intervention and long term follow up. This is possible but unlikely - can you please check and confirm?

Thank you for pointing it out to us, this is indeed a mistake. We have changed Supplementary Table 3 accordingly.

2) "We have changed the sentence to the following: "BLT and DRLT both reduced depressive symptoms in pregnant women with depression."

-Again I must disagree. It is certainly possible that both therapies work, but the authors acknowledge (and reference) in the discussion that the change may be due to several other causes ("the improvement in both groups can be explained by non-specific treatment effects such as the structure offered by the study[....] symptoms decrease related to the course of pregnancy, spontaneous remission, or regression to the mean"). It is very fair to say that symptoms reduced in both groups, but please avoid describing this causal - we simply don't know if it is this or other things (or a mixture).

We have changed the sentence in the abstract to the following (line 40 in the manuscript with track changes): "Depressive symptoms of pregnant women with depression improved in both treatment arms."

We have checked the entire manuscript for these incorrect causal formulations. We have changed the conclusion section in the discussion to the following (lines 477-479 in the manuscript with track changes): "In the present study, depressive symptoms of pregnant women with depression improved in both treatment arms after 6 weeks of treatment."

3) Lastly, the details of the sample size need to be included in the text as well as in the authors response. I am still unable to reproduce the sample size using GLIMMSE (which appears to ask for the mean at each time point), although using a similar method in other software suggests that 63

per arm is about right.

We have added the details of the power calculation to the text of the manuscript, including our mean baseline and end value. Original power calculation were performed using GLIMPSE 2.1.5. This version is no longer available, we include the script to be used in GLIMPSEv3.0.0. In this software version, following the same protocol, the sample size needed is estimated at 122.

In the text of the manuscript we have added the following (lines 135-142 in the manuscript with track changes): "A sample size calculation was performed using GLIMPSE 2.1.5. software 58, with the following parameters: alpha 0.05; beta 0.80; 6 time assessments (continuous, equally spaced); primary test: time*treatment interaction; Sigh-SAD scores assumed at baseline: M:28.0 and SD:7.0, with a linear decrease in symptom scores up to a mean score of 24.0 in the BLT condition. No symptom change was assumed for the DRLT condition; Hotelling-Lawley Trace correction; base correlation 0.4; decay rate 0.05; no additional scaling factors included. To demonstrate this a total sample size of 126 participants, 63 per arm was needed."

VERSION 4 – REVIEW

REVIEWER	Mike Bradburn University of Sheffield
REVIEW RETURNED	01-Oct-2020
GENERAL COMMENTS	My previous comments have been addressed. I thank the authors for considering these and congratulate them on delivering this trial.